# Invasive Fungal Infection Caused by *Magnusiomyces capitatus* in an Immunocompromised Pediatric Patient with Acute Lymphoblastic Leukemia in Mexico City: A Case Report

**DOI:** 10.3390/jof8080851

**Published:** 2022-08-15

**Authors:** Jossue Ortiz-Álvarez, Jesús Reséndiz-Sánchez, Margarita Juárez-Montiel, Juan Alfredo Hernández-García, Edwin Vázquez-Guerrero, César Hernández-Rodríguez, Lourdes Villa-Tanaca

**Affiliations:** 1Laboratorio de Biología Molecular de Bacterias y Levaduras, Departamento de Microbiología, Escuela Nacional de Ciencias Biológicas Instituto Politécnico Nacional, Plan de Ayala y Prolongación Carpio, Santo Tomas, Mexico City 11340, Mexico; 2Hospital Infantil de México, Federico Gómez (HIMFG), Doctor Márquez 162, Doctores, Mexico City 06720, Mexico; 3Laboratorio de Genética Microbiana, Departamento de Microbiología, Escuela Nacional de Ciencias Biológicas Instituto Politécnico Nacional, Plan de Ayala y Prolongación Carpio, Santo Tomas, Mexico City 11340, Mexico

**Keywords:** *Magnusiomyces capitatus*, *Geotrichum capitatum*, *Saprochaete capitata*, invasive fungal infection (IFI), pediatric acute lymphoblastic leukemia (ALL), ITS and 28S phylogeny, MALDI-TOF MS

## Abstract

*Magnusiomyces capitatus* (also denominated “*Geotrichum capitatum*” and “the teleomorph stage of *Saprochaete capitata*”) mainly affects immunocompromised patients with hematological malignancies in rare cases of invasive fungal infections (IFIs). Few cases have been reported for pediatric patients with acute lymphoblastic leukemia (ALL), in part because conventional diagnostic methods do not consistently detect *M. capitatus* in infections. The current contribution describes a systemic infection in a 15-year-old female diagnosed with ALL. She arrived at the Children’s Hospital of Mexico City with a fever and neutropenia and developed symptoms of septic shock 4 days later. *M. capitatus* ENCB-HI-834, the causal agent, was isolated from the patient’s blood, urine, bile, and peritoneal fluid samples. It was identified with matrix-assisted laser desorption/ionization time-of-flight mass spectrometry (MALDI-TOF MS) and a phylogenetic reconstruction using internal transcribed spacer (ITS) and 28S ribosomal sequences. The phylogenetic sequence of *M. capitatus* ENCB-HI-834 clustered with other *M. capitatus*-type strains with a 100% identity. In vitro antifungal testing, conducted with the Sensititre YeastOne susceptibility system, found the following minimum inhibitory concentration (MIC) values (μg/mL): posaconazole 0.25, amphotericin B 1.0, fluconazole > 8.0, itraconazole 0.25, ketoconazole 0.5, 5-flucytosine ≤ 0.06, voriconazole 0.25, and caspofungin > 16.0. No clinical breakpoints have been defined for *M. capitatus*. This is the first clinical case reported in Mexico of an IFI caused by *M. capitatus* in a pediatric patient with ALL. It emphasizes the importance of close monitoring for a timely and accurate diagnosis of neutropenia-related IFIs to determine the proper treatment with antibiotics, antifungals, and chemotherapy for instance including children with ALL.

## 1. Introduction

*Magnusiomyces capitatus* (also denominated “*Geotrichum capitatum*” and “the teleomorph stage of *Saprochaete capitata*”) is a dimorphic yeast commonly isolated from environmental samples, such as wood and feces. It is capable of establishing a symbiotic relationship in the gut microbiota, skin, and respiratory tract of immunocompetent patients [1,2,3]. Recently, *M. capitatus* and *S. capitata* were accepted as synonyms [4].

*M. capitatus* is rarely found as a causal agent of fungemia and invasive fungal infection (IFI). It mainly occurs in immunocompromised patients with hematological malignancies, especially those with neutropenia and acute myeloid leukemia, and results in 90% mortality [5]. *M. capitatus* has sometimes been recognized by conventional methods, including microscopical observation, MALDI-TOF-MS, biochemical tests, and even the β-1-3-d-glucan in vitro assay for early detection [6,7,8,9]. However, a consistent, timely, and accurate identification based on analyzing the phenotypic traits of *M. capitatus* is difficult because it is closely related to other fungi such as *Saprochaete clavata* [10].

## 2. Case Presentation

The current case report concerns the development of IFI stemming from *M. capitatus* in a 15-year-old female diagnosed with acute lymphoblastic leukemia (ALL). The patient was hospitalized on 12 February 2018 with fever and chemotherapy-associated severe neutropenia. Lab tests revealed a low absolute neutrophil count (100 cells/μL), and *Klebsiella pneumoniae* was isolated from a blood culture. Thus, the patient was treated with amikacin (15 mg/kg/day) and cefepime (150 mg/kg/day).

Due to exhibiting septic shock on day 4 of hospitalization, the patient developed symptoms consistent with septic shock and she was subjected to orotracheal intubation and given an additional treatment based on administering amines via a central venous catheter (CVC) as well as a new antibiotic cocktail composed of meropenem (100 mg/kg/day) and vancomycin (40 mg/kg/day). Nevertheless, the signs of septic shock continued for the following 24 h. The patient showed signs of peritonitis, possibly associated with an acute episode of cholecystitis, suggesting a systemic microbial infection. Since fungal structures were found in a blood culture, liposomal amphotericin B (5 mg/kg/day) was administered.

Bile and peritoneal fluid samples were obtained by cholecystostomy and treated with 20% KOH for observation by microscopy, which led to the discovery of hyaline septate macrosiphoned hyphae in the fluid samples (Figure 1A). Antifungal therapy was improved by administering voriconazole (6 mg/kg/day). An abdominal computed tomography (CT) scan of the coronal plane displayed multiple parenchymal injuries in the kidneys and spleen (Figure 1B). At the same time, the eyes were involved in the fungal invasion, and fungal structures were detected in a bone marrow culture. The patient persisted with neutropenia for another 11 days. On day 12, the patient manifested signs of multi-organ failure: the dysfunctions involved the liver, kidneys, lungs, and heart, as well as disseminated intravascular coagulation. The definitive diagnosis of systemic fungal infection was established. Unfortunately, the patient died on day 22 of hospitalization. There was no evidence of fungal structures in the post-mortem cultures.

The fungal pathogen strain ENCB-HI-834 was identified by the following protocol. Pure cultures were obtained and incubated on Sabouraud-dextrose-agar (SDA) at 37 °C and 28 °C for 3–5 days, then stored in 20% glycerol (*v/v*) at −70 °C to await further processing. The presumptive determination of microbes was made by matrix-assisted laser desorption/ionization time-of-flight mass spectrometry (MALDI-TOF MS) with a VITEK^®^ 74 MS & AST VITEK2 apparatus, operated in accordance with the manufacturer’s instructions. The presence of the fungi was confirmed by phylogenetic reconstruction based on the internal transcribed spacer (ITS) and 28S region of LSU rDNA. These gene regions were amplified with the ITS1-ITS4 and LR07-LR7 primers, employing protocols for total DNA extraction and PCR amplifications previously described [11,12,13]. The PCR products were purified with the Clean & Concentrator^TM^ kit (Zymo Research, Orange Co., Irvine, CA, USA) and sequenced by Sanger dideoxy sequencing (Macrogen^®^, Seoul, Korea). Sequences underwent BLASTn analysis by using a nonredundant GenBank library [14]. The multiple alignment of the collected sequences was generated with MUSCLE version 3.8.31 [15].

Maximum-likelihood phylogenetic trees were constructed with nucleotide sequences from the ITS and 28S rRNA gene markers, utilizing MEGAX software, the bootstrap test of 1000 replicates, and the Jukes-Cantor 69 (JC69) nucleotide substitution model computed on jModelTest version 2.1.10 [16,17]. The sequences were deposited in the GenBank with the accession numbers MN832904 (ITS) and MN833644 (28S). A phylogenetic reconstruction confirmed the identity of the fungal strain ENCB-HI-834 as *M. capitatus*, which clustered with other *M. capitatus* strains with a 100% identity (Figure 2A,B; Appendix A). At the same time, antifungal testing was carried out with the Sensititre YeastOne^®^ susceptibility system (Thermo Fisher Scientific^TM^, Waltham, MA, USA), a colorimetric broth microdilution method, as indicated by the manufacturer. The resulting minimum inhibitory concentration (MIC) values were as follows: posaconazole): posaconazole 0.25, amphotericin B 1.0, fluconazole > 8.0, itraconazole 0.25, ketoconazole 0.5, 5-flucytosine ≤ 0.06, voriconazole 0.25, and caspofungin > 16.0. No clinical breakpoints have been defined for *Magnusiomyces* spp. *(**Geotrichum* spp. or *Saprochaete* spp.).

## 3. Discussion

Infective complications have been documented during the treatment of children with ALL [18]. *Aspergillus*, *Candida*, *Cryptococcus*, and *Pneumocystis* have been the most frequently identified fungal pathogens in immunocompromised patients [10]. However, fungal species other than those mentioned above have emerged in recent years, some resistant to conventional antifungals as causal agents of IFIs, such as *M. capitatus* [10]. Since the latter can lead to bloodstream infections and invasive and disseminated multi-organ disease, a timely diagnosis and treatment is a critical challenge [4]. Most patients with ALL who develop invasive *Magnusiomyces* infection, fungemia, and IFI do so due to profound neutropenia secondary to chemotherapy and after dexamethasone treatment [10,18].

*M. capitatus*, recently defined as an emerging pathogen, is mostly found in patients with hematological malignancies, particularly acute leukemia. For instance, it was detected in an adult male diagnosed with ALL and hospitalized in Mexico City’s General Hospital. IFIs in children with ALL have been scantly reported. The current contribution details a case of systemic fungal infection by *M. capitatus* ENCB-HI-834 in a patient admitted to the Children’s Hospital of Mexico City. Systemic infections caused by *M. capitatus* have been mainly documented in the Mediterranean countries of Europe, including Italy, France, and Spain [1,19,20]. They have also been reported in other areas of Europe (Slovakia, Switzerland, and the Czech Republic) [4] and in Asia (India, Nepal, Kuwait, and China) [21,22,23]. In the Americas, *M. capitatus* was recently isolated from an alcoholic male patient in the United States [24].

The conventional methods for recognizing *M. capitatus* as the agent of an IFI have not provided consistently accurate results. For example, *S. clavata* is frequently confused with *S. capitata* and vice versa because they are identical macroscopically and microscopically [10]. Therefore, phylogenetic reconstruction methods were employed for the characterization of the fungal strain. Molecular identification and phylogeny have been used previously for the detection of *M. capitatus* and its differentiation from closely related species such as *S. clavata* [24,25,26]. These techniques were instrumental for recognizing *M. capitatus* ENCB-HI-834 as the agent of the present systemic infection. The examination of samples by the molecular sequences of ITS and 18S can probably help to establish the relationship of *M. capitatus* strains from various geographical sites. However, we suggest that phylogenomic studies derived from whole-genome sequencing (WGS) are advantageous [27]. A complete genome analysis of *M. capitatus* will likely allow for a phylogenomic and pan-genomic evaluation of the strains isolated from patients [27], as well as provide insights into virulence factors, sensitivity to antifungals, resistance-associated mutations, new therapeutic targets, new treatments, the relationship between geographic clades, and the relationship between environmental strains and opportunistic pathogen strains (https://www.cdc.gov/fungal/outbreaks/wgs.html, accessed on 9 July 2022).

Given that no clinical breakpoints have been defined for *Saprochaete* and related species, antifungal susceptibility data should be interpreted with caution based on experience with the protocols established for the administration of antifungals [10]. According to in vitro susceptibility results from numerous case reports, *M. capitatus* is intrinsically resistant to echinocandins and highly resistant to fluconazole. Liposomal amphotericin B appears to be the antimicrobial of choice for empirically treating invasive *M. capitatus* [10], but the combined administration of liposomal amphotericin B with voriconazole was found to be successful in the treatment of a 6-year-old pediatric patient with neutropenia, infected with *Magnusiomyces clavatus* [28]. To date, there is no established antifungal treatment therapeutic for invasive *Magnusiomyces* spp. infections, mainly due to the low frequency with which this species appears, its difficult diagnosis by conventional methods, and the lack of antifungal breakpoints [3]. Therefore, treatment recommendations for *M. capitatus* are based on the therapeutic experience of experts and data from the few reported cases [10].

Unfortunately, infection with *M. capitatus* is associated with a poor prognosis, and IFIs stemming from this species have a high mortality rate. For patients with neutropenia, the prognosis is even worse. Even when the treatment uses liposomal amphotericin B, itraconazole, and flucytosine, the mortality rate remains as high as 50–90% [18]. Thus, early antifungal treatment could be a critical factor for reducing mortality. According to observations of children with ALL, infection is the leading cause of treatment-related mortality. Infection is more likely during the induction phase, and the risk of infection-related mortality is greater in children with Down syndrome during all phases of treatment [29].

The emergence of new species or rare species of fungi as pathogens of humans capable of causing IFIs have several possible explanations: (a) opportunistic environmental fungi could exist in close living environments and cause fungal diseases in individuals with compromised health, but due to environmental conditions, they have been able to evolve and allow for the selection of fungi resistant to antifungals; (b) conventional techniques did not allow for the precise identification of fungal species, unlike current molecular techniques; (c) the conditions of the patients have changed, and now there are significantly more immunocompromised patients; (d) treatments with increasingly potent antimicrobials and co-infections with other viral or bacterial microorganisms have favored the establishment of polymicrobial-infection cases [30,31]. The relatively recent appearance and identification of *M. capitatus* may be due to some of the aforementioned factors.

## 4. Conclusions

This study represents the first clinical case reported in Mexico of an IFI caused by *M. capitatus* in an adolescent with ALL. ITS marker and MALDI-TOF MS were invaluable tools for identifying the species as *M. capitatus*. The report of this case provides epidemiological information. It may draw attention to the study of the virulence factors and their mechanisms of antifungal resistance of this species of emerging fungus, the application of alternative treatment protocols with known antibiotics and antifungals, as well as the design of new antifungals that help to solve the problem of multi-resistance to conventional antifungals. The current case emphasizes the critical importance of close monitoring for the timely and accurate diagnosis of neutropenia-related IFIs to determine the proper treatment with antibiotics, antifungals, and chemotherapy, such as in cases involving pediatric patients with ALL.

## Figures and Tables

**Figure 1 jof-08-00851-f001:**
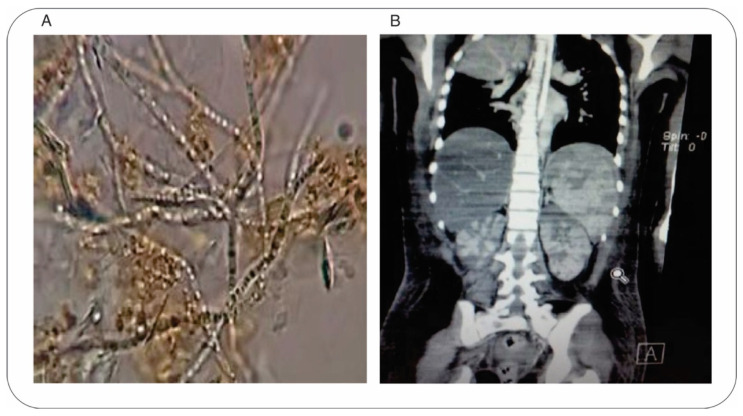
Identification of *M. capitatus* as the causal agent of IFI. (**A**) Microscopic evaluation (Zeiss Primo Star 415500 optical microscope, 40×) of a fresh sample of bile fluid with 20% of KOH, observing hyaline septate macrosiphoned hyphae structures. (**B**) Detection of *M. capitatus* invasion of internal organs with an abdominal CT scan. The coronal plane shows multiple injuries in the spleen and kidneys.

**Figure 2 jof-08-00851-f002:**
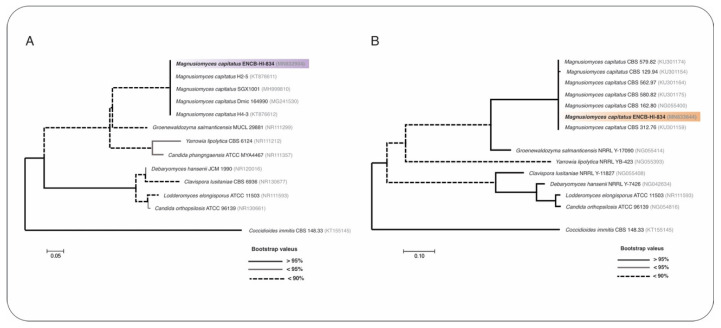
Molecular identification of *M. capitatus* ENCB-HI-834 was accomplished by using maximum-likelihood phylogenetic reconstruction, performed with sequences from two molecular markers: (**A**) internal transcribed spacer (ITS) and (**B**) 28S. The Jukes-Cantor 69 (JC69) nucleotide substitution test was employed for the generation of phylogenetic trees. The style of the branches represent the Bootstrap values computed with 1000 replicates. Branch lengths are proportional to the number of substitutions per site (see scale bar). The ITS (mauve color) and 28S (orange color) sequences correspond to *M. capitatus* ENCB-HI-834. *Coccidioides immitis*-type strains served as external groups.

## Data Availability

The internal transcribed spacer 1 and 2 of the *M. capitatus* strain ENCB-HI-834 (partial sequences) and the 5.8S ribosomal RNA gene (complete sequence) were registered in GenBank (MN832904.1). The large subunit ribosomal RNA gene of the *M. capitatus* strain ENCB-HI-834 (partial sequence) can be found in GenBank (MN833644.1).

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
