# Peer review of "Invasive Fungal Infection Caused by Magnusiomyces capitatus in an Immunocompromised Pediatric Patient with Acute Lymphoblastic Leukemia in Mexico City: A Case Report"

_jof, 2022, doi:10.3390/jof8080851_

Round 1

Reviewer 1 Report

I don't understand how you present the susceptibility results with the Sensititre YeastOne: what do you mean with voriconazole ≥ 0.25 or itraconazole ≥0.25 and caspofungin ≤16? Is that a typo with the ≥ and ≤ sign?

Some of your sentences seem copied and pasted from your Reference 10.

Your statement "Liposomal amphotericin B appears to 166 be the antimicrobial of choice for empirically treating invasive M. capitatus ENCB-HI-834." is not supported by the references you provide. 

The "Conclusions" section is unconvincing. I fail to see why your report is significant. And the statement "The current case emphasizes the critical importance of close monitoring for the timely and accurate diagnosis of neutropenia-related IFIs to 184 determine the proper treatment with antibiotics, antifungals, and chemotherapy, such as 185 in pediatric patients with ALLmay not be true, since the "close monitoring" and the "proper treatment" did not result in a favorble outcome.

Author Response

Response to Reviewer 1

First of all, I want to thank the reviewers for all their comments on this manuscript, which will undoubtedly improve its quality and better understanding for its readers.

All changes in the new version of the manuscript are highlighted in yellow.

Reviewer #1

Principio del formulario

  1. I don't understand how you present the susceptibility results with the Sensititre YeastOne: what do you mean with voriconazole ≥ 0.25 or itraconazole ≥0.25 and caspofungin ≤16? Is that a typo with the ≥ and ≤ sign?

Response: Thank you for this observation; the signs have been corrected as follows. The definition of minimum inhibitory concentration (MIC) is the minimum concentration tested at which fungal growth is not observed. However, when growth is observed in all wells, even at the highest concentration, it should be reported with > sign (greater than). On the other hand, when there is growth inhibition in all the concentrations tested, since it cannot be asserted that the minimum concentration tested is the MIC, it is reported with ≤ sign (less than or equal to) the lowest concentration tested. The signs have been corrected.

Lines: 34-35 and 115-117

In vitro antifungal testing, conducted with the Sensititre YeastOne susceptibility system, found the following minimum inhibitory concentration (MIC) values (mg/mL): posaconazole 0.25, amphotericin B 1.0, fluconazole >8.0, itraconazole 0.25, ketoconazole 0.5, 5-flucytosine ≤0.06, voriconazole 0.25, and caspofungin 16.0.

  1. Some of your sentences seem copied and pasted from your Reference 10.

Response: Reference [10]  is significant for discussing this case report. An analysis of plagiarism was carried out in the "Grammarly" application, and, eliminating the bibliographical references, it yielded a result of 7% similarity with other texts searched on the Internet. And only two of the sentences coincided with the article in El Zein (reference 10). Therefore, these sentences were redrafted:

  1. A) New version of the paragraph, Line 128-136

Infective complications have been documented during treating children with ALL [18]. Aspergillus, Candida, Cryptococcus, and Pneumocystis have been the most frequently identified fungal pathogens in immunocompromised patients [10]. However, fungal species other than those mentioned above have emerged in recent years, some resistant to conventional antifungals as causal agents of IFIs, such as M. capitatus [10]. Since the latter can lead to bloodstream infections and invasive and disseminated multi-organ disease, timely diagnosis and treatment is a critical challenge [4]. Most patients with ALL who develop invasive Magnusiomyces infection, fungemia, and IFI are due to profound neutropenia secondary to chemotherapy and after dexamethasone treatment [10, 18].

  1. B) New version of the paragraph, Lines 174-178

To date, there is no established antifungal treatment therapeutic for invasive Magnusiomyces spp infections, mainly due to the low frequency with which this species appears, its difficult diagnosis by conventional methods, and the lack of antifungal breakpoints [3]. Therefore, treatment recommendations for M. capitatus are based on the therapeutic experience of experts and data in reports of the few reported cases [10].

  1. Your statement "Liposomal amphotericin B appears to 166 be the antimicrobial of choice for empirically treating invasive M. capitatus ENCB-HI-834." is not supported by the references you provide. 

Response (Lines 170-174):

Thanks for the comment; this new version cites the appropriate reference that supports empiric treatment with liposomal amphotericin B. In addition to citing reference 10, we added reference 28 (Leoni et al., 2018)

Liposomal amphotericin B appears to be the antimicrobial of choice for empirically treating invasive M. capitatus  [10], but the combined administration of liposomal amphotericin B with voriconazole was found to be a successful treatment in a 6-year-old pediatric patient with neutropenia, infected with Magnusiomyces clavatus [28].

This reference was included in the corresponding section

  1. Leoni M.; Riccardi N.; Rotulo G.A.; Godano E.; Faraci M.; Bandettini R.; Esposto M.C.; Castagnola E. Magnusiomyces clavatus infection in a child after allogeneic hematotopoietic stem cell transplantation: Diagnostic and therapeutic implications. Med. Mycol. Case Rep. 2018 21, 65-67.

  1. The "Conclusions" section is unconvincing. I fail to see why your report is significant. And the statement "The current case emphasizes the critical importance of close monitoring for the timely and accurate diagnosis of neutropenia-related IFIs to 184 determine the proper treatment with antibiotics, antifungals, and chemotherapy, such as 185 in pediatric patients with ALL "may not be true, since the "close monitoring" and the "proper treatment" did not result in a favorable outcome.

Response. Thank you for this observation. The reports of clinical cases of IFIs related to fungi of rare species are essential for several reasons: they provide epidemiological information, they encourage the identification of the fungus to the species using unconventional techniques and motivate the study of the virulence factors of these emerging fungi and their mechanisms of antifungal resistance, as well as the search and design of new antifungals that help solve the problem of multi-resistance to conventional antifungals. In addition, the study of the appearance of these fungi in intrahospital environments could help establish protocols to prevent them from becoming healthcare-associated infections (HAIs) and to discern whether it is an HAI infection or a community-associated infection (CAI). However, this report has many edges, and the reading of this manuscript will depend on the reader's specialty.

The conclusions have been redrafted to be convincing (lines 202-212):

This study represents the first clinical case reported in Mexico of an IFI caused by M. capitatus in an adolescent with ALL. ITS marker and MALDI-TOF MS were invaluable tools for identifying the species as M. capitatus. The report of this case provides epidemiological information. It may draw attention to the study of the virulence factors and their mechanisms of antifungal resistance of this species of emerging fungus, the application of alternative treatment protocols with known antibiotics and antifungals, as well as the design of new antifungals that help to solve the problem of multi-resistance to conventional antifungals. The current case emphasizes the critical importance of close monitoring for the timely and accurate diagnosis of neutropenia-related IFIs to determine the proper treatment with antibiotics, antifungals, and chemotherapy, such as in pediatric patients with ALL.

Response to Reviewer 1

First of all, I want to thank the reviewers for all the comments they have made to this manuscript, which will undoubtedly improve its quality and better understanding for its readers.

All changes in the new version of the manuscript are highlighted in yellow.

Reviewer #1

Principio del formulario

  1. I don't understand how you present the susceptibility results with the Sensititre YeastOne: what do you mean with voriconazole ≥ 0.25 or itraconazole ≥0.25 and caspofungin ≤16? Is that a typo with the ≥ and ≤ sign?

Response: Thank you for this observation; the signs have been corrected as follows. The definition of minimum inhibitory concentration (MIC) is the minimum concentration tested at which fungal growth is not observed. However, when growth is observed in all wells, even at the highest concentration, it should be reported with > sign (greater than). On the other hand, when there is growth inhibition in all the concentrations tested, since it cannot be asserted that the minimum concentration tested is the MIC, it is reported with ≤ sign (less than or equal to) the lowest concentration tested. The signs have been corrected.

Lines: 34-35 and 115-117

In vitro antifungal testing, conducted with the Sensititre YeastOne susceptibility system, found the following minimum inhibitory concentration (MIC) values (mg/mL): posaconazole 0.25, amphotericin B 1.0, fluconazole >8.0, itraconazole 0.25, ketoconazole 0.5, 5-flucytosine ≤0.06, voriconazole 0.25, and caspofungin 16.0.

  1. Some of your sentences seem copied and pasted from your Reference 10.

Response: Reference [10]  is significant for discussing this case report. An analysis of plagiarism was carried out in the "Grammarly" application, and, eliminating the bibliographical references, it yielded a result of 7% similarity with other texts searched on the Internet. And only two of the sentences coincided with the article in El Zein (reference 10). Therefore, these sentences were redrafted:

  1. A) New version of the paragraph, Line 128-136

Infective complications have been documented during treating children with ALL [18]. Aspergillus, Candida, Cryptococcus, and Pneumocystis have been the most frequently identified fungal pathogens in immunocompromised patients [10]. However, fungal species other than those mentioned above have emerged in recent years, some resistant to conventional antifungals as causal agents of IFIs, such as M. capitatus [10]. Since the latter can lead to bloodstream infections and invasive and disseminated multi-organ disease, timely diagnosis and treatment is a critical challenge [4]. Most patients with ALL who develop invasive Magnusiomyces infection, fungemia, and IFI are due to profound neutropenia secondary to chemotherapy and after dexamethasone treatment [10, 18].

  1. B) New version of the paragraph, Lines 174-178

To date, there is no established antifungal treatment therapeutic for invasive Magnusiomyces spp infections, mainly due to the low frequency with which this species appears, its difficult diagnosis by conventional methods, and the lack of antifungal breakpoints [3]. Therefore, treatment recommendations for M. capitatus are based on the therapeutic experience of experts and data in reports of the few reported cases [10].

  1. Your statement "Liposomal amphotericin B appears to 166 be the antimicrobial of choice for empirically treating invasive M. capitatus ENCB-HI-834." is not supported by the references you provide. 

Response (Lines 170-174):

Thanks for the comment; this new version cites the appropriate reference that supports empiric treatment with liposomal amphotericin B. In addition to citing reference 10, we added reference 28 (Leoni et al., 2018)

Liposomal amphotericin B appears to be the antimicrobial of choice for empirically treating invasive M. capitatus  [10], but the combined administration of liposomal amphotericin B with voriconazole was found to be a successful treatment in a 6-year-old pediatric patient with neutropenia, infected with Magnusiomyces clavatus [28].

This reference was included in the corresponding section

  1. Leoni M.; Riccardi N.; Rotulo G.A.; Godano E.; Faraci M.; Bandettini R.; Esposto M.C.; Castagnola E. Magnusiomyces clavatus infection in a child after allogeneic hematotopoietic stem cell transplantation: Diagnostic and therapeutic implications. Med. Mycol. Case Rep. 2018 21, 65-67.

  1. The "Conclusions" section is unconvincing. I fail to see why your report is significant. And the statement "The current case emphasizes the critical importance of close monitoring for the timely and accurate diagnosis of neutropenia-related IFIs to 184 determine the proper treatment with antibiotics, antifungals, and chemotherapy, such as 185 in pediatric patients with ALL "may not be true, since the "close monitoring" and the "proper treatment" did not result in a favorable outcome.

Response. Thank you for this observation. The reports of clinical cases of IFIs related to fungi of rare species are essential for several reasons: they provide epidemiological information, they encourage the identification of the fungus to the species using unconventional techniques and motivate the study of the virulence factors of these emerging fungi and their mechanisms of antifungal resistance, as well as the search and design of new antifungals that help solve the problem of multi-resistance to conventional antifungals. In addition, the study of the appearance of these fungi in intrahospital environments could help establish protocols to prevent them from becoming healthcare-associated infections (HAIs) and to discern whether it is an HAI infection or a community-associated infection (CAI). However, this report has many edges, and the reading of this manuscript will depend on the reader's specialty.

The conclusions have been redrafted to be convincing (lines 202-212):

This study represents the first clinical case reported in Mexico of an IFI caused by M. capitatus in an adolescent with ALL. ITS marker and MALDI-TOF MS were invaluable tools for identifying the species as M. capitatus. The report of this case provides epidemiological information. It may draw attention to the study of the virulence factors and their mechanisms of antifungal resistance of this species of emerging fungus, the application of alternative treatment protocols with known antibiotics and antifungals, as well as the design of new antifungals that help to solve the problem of multi-resistance to conventional antifungals. The current case emphasizes the critical importance of close monitoring for the timely and accurate diagnosis of neutropenia-related IFIs to determine the proper treatment with antibiotics, antifungals, and chemotherapy, such as in pediatric patients with ALL.

Reviewer 2 Report

Excellent and interesting work. 

Just few issues to clarify:

1. as reported by the Journal guidelines:

  • "Informed Consent Statement: Any research article describing a study involving humans should contain this statement. Please add “Informed consent was obtained from all subjects involved in the study.” OR “Patient consent was waived due to REASON (please provide a detailed justification).” OR “Not applicable” for studies not involving humans. You might also choose to exclude this statement if the study did not involve humans.
  • Written informed consent for publication must be obtained from participating patients who can be identified (including by the patients themselves). Please state “Written informed consent has been obtained from the patient(s) to publish this paper” if applicable."

In this case, the authors reported "Informed Consent Statement: Not applicable ". Please, be sure it is the case. Otherwise, provide an adequate statement.

2. in the case description, it was unclear if the fungal identification came DURING the hospitalization and at what time, or if it was post-mortem. Considering the tentative therapy with amphotericin B, it seems that M.capitatus was approximately identified on day five.

3. since the authors at the beginning have reported Klebsiella, they could better explain the case if the fungal infection was already present from the beginning but not yet identified (as. I suppose) or was a consequence of the massive antimicrobial drugs in such fragile patient. 

4. It could be interesting to add a few lines in the discussion section about the following point:

- When authors wrote (lines 128-9) "Over the last few years, however, less common fungal species have emerged as causative agents 129 of IFIs, such as M. capitatus [10] ". Why are they emerging more during last year? For epidemiological/ecological reasons, or because the novel and sensitive technique allow to identify it when other conventional tools fail?

Author Response

  1. Response to Reviewer 2

Reviewer #2

Excellent and interesting work. 

Just few issues to clarify:

  1. As reported by the Journal guidelines:
  • "Informed Consent Statement: Any research article describing a study involving humans should contain this statement. Please add “Informed consent was obtained from all subjects involved in the study.” OR “Patient consent was waived due to REASON (please provide a detailed justification).” OR “Not applicable” for studies not involving humans. You might also choose to exclude this statement if the study did not involve humans.
  • Written informed consent for publication must be obtained from participating patients who can be identified (including by the patients themselves). Please state “Written informed consent has been obtained from the patient(s) to publish this paper” if applicable."

In this case, the authors reported "Informed Consent Statement: Not applicable ". Please, be sure it is the case. Otherwise, provide an adequate statement.

Response. The Informed Consent Statement is "Not Applicable" because no experiments were performed on the patient, and no new treatment protocols were administered. All treatments applied to the pediatric patient were within the established protocols. The application of Liposomal Amphotericin B is discussed in answer to the next question. The isolated strain is part of the laboratory's collection of fungal strains, and the identification was made post-mortem.

  1. In the case description, it was unclear if the fungal identification came DURING the hospitalization and at what time, or if it was post-mortem. Considering the tentative therapy with amphotericin B, it seems that M. capitatus was approximately identified on day five.

Response. Amphotericin B therapy was decided on the fifth day when the fungal structures were observed microscopically. Indeed, the identification of the fungus was carried out post-mortem. As mentioned in the results and discussion section, there is no specific protocol for M. capitatus and other related species, however, Amphotericin B is the broad-spectrum antifungal of choice selected for these cases.

  1. Since the authors at the beginning have reported Klebsiella, they could better explain the case if the fungal infection was already present from the beginning but not yet identified (as. I suppose) or was a consequence of the massive antimicrobial drugs in such fragile patient. 

Response. The reviewer's assumption is highly probable; it is possible that the fungus was present from the beginning and that as a consequence of the administration of antimicrobials, an imbalance in the patient's microbiota (dysbiosis) resulted, and therefore the fungus could grow and be detected on the fifth day.

  1. It could be interesting to add a few lines in the discussion section about the following point:

- When authors wrote (lines 128-9) "Over the last few years, however, less common fungal species have emerged as causative agents 129 of IFIs, such as M. capitatus [10] ". Why are they emerging more during last year? For epidemiological/ecological reasons, or because the novel and sensitive technique allow to identify it when other conventional tools fail?

Response: This is an excellent point to discuss; thanks for the suggestion. The following lines were added to the discussion:

Lines 190-199

The emergence of new species or rare species of fungi as pathogens of humans capable of causing IFIs can have several explanations: a) There are already known environmental fungal species, but due to environmental conditions, they have been able to evolve and allow the selection of fungi resistant to antifungals; b) Conventional techniques did not allow the precise identification of fungal species, and current molecular techniques do; c) The conditions of the patients have changed, and now there are a more significant number of immunocompromised patients; d) Treatments with increasingly potent antimicrobials and co-infections with other viral or bacterial microorganisms have favored the establishment of polymicrobial infections [30,31]. The relatively recent appearance and identification of M. capitatus may be due to some of the aforementioned factors.

References included:

  1. Fisher, M.C., Alastruey-Izquierdo, A., Berman, J. et al.Tackling the emerging threat of antifungal resistance to human health. Nat. Rev. Microbiol.2022, 29, 1-15

  1. Wiederhold N.P. Emerging Fungal Infections: New Species, New Names, and Antifungal Resistance. Clin. Chem. 2021, 68, 83-90.

Reviewer 3 Report

Dear Authors,

The manuscript ID: jof-1835709 entitled Invasive fungal infection caused by Magnusiomyces capitatus in an immunocompromised pediatric patient with acute lymphoblastic leukemia in Mexico City: A case report” written by Jossue Ortiz-Álvarez, Jesús Reséndiz, Margarita Juárez-Montiel, Juan Alfredo Hernández-García, Edwin Vázquez-Guerrero, César Hernández-Rodríguez and Lourdes Villa-Tanaca is very original.

Invasive fungal infection is one of the most common nosocomial bloodstream infections and may result in invasive diseases in immunocompromised patients. In recent years, the incidence of IFI has been increasing, especially within hospitalized patients. The pathogens responsible for IFI are mostly opportunistic pathogens, including Candida, Aspergillus, and Cryptococcus. In turn, Geotrichum capitatum infection has a very low incidence rate with atypical clinical symptoms, making diagnosis difficult, and it has a poor prognosis.

This case report describes a systemic infection in a 15-year-old female diagnosed with acute lymphoblastic leukemia. Unfortunately, the patient died on day 22 of hospitalization. It emphasizes the importance of close monitoring for a timely and accurate diagnosis of neutropenia-related IFIs to determine the proper treatment with antibiotics, antifungals, and chemotherapy. The whole manuscript is properly organized and well written.

I have only small suggestions in order to improve paper, which are the following:

Lines 34, 114: 5 FC ≥0.06 – please provide the full name of the antimycotic.

In my opinion, this manuscript is valuable and may be accepted for the publication in “Journal of Fungi”.

With highest regards,

Author Response

  1. Response to Reviewer 3

First of all, I want to thank the reviewers for all the comments they have made to this manuscript, which will undoubtedly improve its quality and better understanding for its readers.

All changes in the new version of the manuscript are highlighted in yellow.

Reviewer #3

Comments and Suggestions for Authors

Dear Authors,

The manuscript ID: jof-1835709 entitled „Invasive fungal infection caused by Magnusiomyces capitatus in an immunocompromised pediatric patient with acute lymphoblastic leukemia in Mexico City: A case report” written by Jossue Ortiz-Álvarez, Jesús Reséndiz, Margarita Juárez-Montiel, Juan Alfredo Hernández-García, Edwin Vázquez-Guerrero, César Hernández-Rodríguez and Lourdes Villa-Tanaca is very original.

Invasive fungal infection is one of the most common nosocomial bloodstream infections and may result in invasive diseases in immunocompromised patients. In recent years, the incidence of IFI has been increasing, especially within hospitalized patients. The pathogens responsible for IFI are mostly opportunistic pathogens, including Candida, Aspergillus, and Cryptococcus. In turn, Geotrichum capitatum infection has a very low incidence rate with atypical clinical symptoms, making diagnosis difficult, and it has a poor prognosis.

This case report describes a systemic infection in a 15-year-old female diagnosed with acute lymphoblastic leukemia. Unfortunately, the patient died on day 22 of hospitalization. It emphasizes the importance of close monitoring for a timely and accurate diagnosis of neutropenia-related IFIs to determine the proper treatment with antibiotics, antifungals, and chemotherapy. The whole manuscript is properly organized and well written.

  1. I have only small suggestions in order to improve paper, which are the following:

Lines 34, 114: 5 FC ≥0.06 – please provide the full name of the antimycotic.

Response: The full name of the antifungal was written (Lines 35 and 116):  5-flucytosine

In my opinion, this manuscript is valuable and may be accepted for the publication in “Journal of Fungi”.

Reviewer 4 Report

The current case report represents a practical contribution to the emerging issue of treatment-related infections and mortality. It describes a rare systemic fungal infection in the context of comorbidities and patient’s personal characteristics. It highlights the problem of new/emerging health conditions such as rare fungal infections. It also provides a great and detailed illustration on the diagnostic process, especially in the phylogenetics field. Systemic involvement and search for microbes on surgical biopsia and biles represent a unique aspect of this report. I also appreciated the search for minimum inhibitory concentration (MIC) that could be useful in the future development of standard breakpoints for M. capitatus.

Table S1 and Figure 2 are adequate methods to compare related species and to study phylogenesis. In fact, a similar representation was found in reference 12. 

I agree that publication would be useful as it would contribute to enlarge the scant knowledge available for this disease, however, I would suggest:

- expanding on the patient's anamnesis and her risk factors (time of diagnosis, immunofenotipization and cytogenetics, regimen of treatment administered and phase of chemiotherapy when the infective complication occurred, other risk factors such as metabolic diseases or the presence of CVC. 

- it could be useful to include the current report in a wider casistic and to offer contributions to future guidelines for antifungal prophylaxis and treatment. An example of categorization of data could be found in Reference N°29 (O’Connor et al., 2014) or the publication at the website link https://www.ncbi.nlm.nih.gov/pmc/articles/PMC8846490/?report=classic (Noster et al., 2022).

- I also suggest the authors start a national dataset on this fungus if they truly have seen more than one case. 

Other minor changes: 

Line 47-48: “It is capable of establishing a symbiotic relationship in the gut microbiota, skin, and respiratory tract of immunocompetent patients”[5].
Line 59: The current case report concerns the development of IFI from M. capitatus in a 15-year-old female diagnosed with acute lymphoblastic leukemia”
Line 61: “The patient was hospitalized on February 12th of 2018 with fever and chemotherapy-associated severe neutropenia.” 

When a measurement unit concerning posology of treatment, using the unit mg/kg/die
Line 65-68: On day 4 of hospitalization the patient developed symptoms suitable with septic shock (maybe specify symptoms) and she was subjected to orotracheal intubation and cardiovascular support was provided with amines and antimicrobial therapy was implemented with meropenem (100 mg/kg/die) and vancomycin (40mg/kg/die).

Line 69: “ The patient showed signs of peritonism, possibly associated with an acute episode of cholecystitis, suggesting a systemic microbial infection.

Line 75: Antifungal therapy was improved administering voriconazole (6 mg/kg/die)

Line 78: At the same time eyes were involved in the fungal invasion.

Line 80-81: On day 12, the patient manifested signs of multi-organ failure: dysfunctions involved liver, kidneys, lungs, and heart and manifested disseminated intravascular coagulation.

Line 127: Infective complications have been documented during children’s treatment for ALL.

Line 135: [...] is mostly found in patients with hematological malignancies, in
particolar acute leukemia.

Reference 19: the Mexican case report mentioned was not available searching on the specific journal website.

Author Response

  1. Response to Reviewer 4

First of all, I want to thank the reviewers for all the comments they have made to this manuscript, which will undoubtedly improve its quality and better understanding for its readers. Many thanks to all the comments that have been made to this manuscript, which will undoubtedly improve its quality and better understanding for its readers.

All changes in the new version of the manuscript are highlighted in yellow

Reviewer #4   The current case report represents a practical contribution to the emerging issue of treatment-related infections and mortality. It describes a rare systemic fungal infection in the context of comorbidities and patient’s personal characteristics. It highlights the problem of new/emerging health conditions such as rare fungal infections. It also provides a great and detailed illustration on the diagnostic process, especially in the phylogenetics field. Systemic involvement and search for microbes on surgical biopsia and biles represent a unique aspect of this report. I also appreciated the search for minimum inhibitory concentration (MIC) that could be useful in the future development of standard breakpoints for M. capitatus.

Table S1 and Figure 2 are adequate methods to compare related species and to study phylogenesis. In fact, a similar representation was found in reference 12. 

I agree that publication would be useful as it would contribute to enlarge the scant knowledge available for this disease, however, I would suggest:

  1. Expanding on the patient's anamnesis and her risk factors (time of diagnosis, immunofenotipization and cytogenetics, regimen of treatment administered and phase of chemiotherapy when the infective complication occurred, other risk factors such as metabolic diseases or the presence of CVC. 

Response. Unfortunately, the patient's clinical history does not mention the chemotherapy phase or the existence of metabolic diseases; however, the presence of a central venous catheter (CVC) was necessary for the administration of amines.

Response (lines 68-69):

Due to exhibiting septic shock on day 4 of hospitalization, the patient was subjected to orotracheal intubation and given an additional treatment based on administering amines via central venous catheter (CVC)

  1. It could be useful to include the current report in a wider casistic and to offer contributions to future guidelines for antifungal prophylaxis and treatment. An example of categorization of data could be found in Reference N°29 (O’Connor et al., 2014) or the publication at the website link https://www.ncbi.nlm.nih.gov/pmc/articles/PMC8846490/?report=classic (Noster et al., 2022).

Response. Thanks for the suggestion. Currently, M. capiatus strain will be deposited in the Mycology Laboratory of the National Institute of Diagnosis and Reference of Mexico (INDRE). Later, if the article is accepted, it will be uploaded to the suggested platforms.

 3.- I also suggest the authors start a national dataset on this fungus if they truly have seen more than one case. 

Response. This suggestion will be addressed via the corresponding mechanisms.

  1. Other minor changes: 

Response: The following minor changes were addressed:

 Line 47-48: “It is capable of establishing a symbiotic relationship in the gut microbiota, skin, and respiratory tract of immunocompetent patients”[5]. (Line 49)

Line 59: The current case report concerns the development of IFI from M. capitatus in a 15-year-old female diagnosed with acute lymphoblastic leukemia”. (Line 60)

Line 61: “The patient was hospitalized on February 12th of 2018 with fever and chemotherapy-associated severe neutropenia.” (Line 62)

When a measurement unit concerning posology of treatment, using the unit mg/kg/die

Line 65-68: On day 4 of hospitalization the patient developed symptoms suitable with septic shock (maybe specify symptoms) and she was subjected to orotracheal intubation and cardiovascular support was provided with amines and antimicrobial therapy was implemented with meropenem (100 mg/kg/die) and vancomycin (40mg/kg/die). (Lines 66-70).

Line 69: “ The patient showed signs of peritonism, possibly associated with an acute episode of cholecystitis, suggesting a systemic microbial infection. (Line 71)

Line 75Antifungal therapy was improved by administering voriconazole (6 mg/kg/die) (Lines 76-77)

Line 78: At the same time eyes were involved in the fungal invasion. (Line 79)

Line 80-81: On day 12, the patient manifested signs of multi-organ failure: dysfunctions involved liver, kidneys, lungs, and heart and manifested disseminated intravascular coagulation. (Lines 81-82)

Line 127Infective complications have been documented during children’s treatment for ALL. (Line 128)

Line 135: [...] is mostly found in patients with hematological malignancies, in

particular acute leukemia. (Line 139)

Reference 19: the Mexican case report mentioned was not available searching on the specific journal website.

Reference 19 was removed.

Round 2

Reviewer 1 Report

Thank you for addressing my comments

Reviewer 4 Report

No further comments, although the lack of more anamnestic details is quite a relevant drawback.